# Learning to Branch with Offline Reinforcement Learning

## Abstract

Mixed Integer Linear Program (MILP) solvers are mostly built upon a branch-and-bound (B&B) algorithm, where the efficiency of traditional solvers heavily depends on hand-crafted heuristics for branching. Such a dependency significantly limits the success of those solvers because such heuristics are often difficult to obtain, and not easy to generalize across domains/problems. Recent deep learning approaches aim to automatically learn the branching strategies in a data-driven manner, which removes the dependency on hand-crafted heuristics but introduces a dependency on the availability of high-quality training data. Obtaining the training data that demonstrates near-optimal branching strategies can be a difficult task itself, especially for large problems where accurate solvers have a hard time scaling and producing near-optimal demonstrations. This paper overcomes this obstacle by proposing a new offline reinforcement learning (RL) approach, namely the *Ranking-Constrained Actor-Critic* algorithm, which can efficiently learn good branching strategies from sub-optimal or inadequate training signals. Our experiments show its advanced performance in both prediction accuracy and computational efficiency over previous methods for different types of MILP problems on multiple evaluation benchmarks.

## 1 Introduction

Combinatorial optimization (CO) has been a fundamental challenge in computer science for decades, with a wide range of real-world applications, including supply chain management, logistics optimization (Chopra & Meindl, 2001), workforce scheduling (Ernst et al., 2004), financial portfolioing (Rubinstein, 2002; Lobo et al., 2007), compiler optimization (Trofin et al., 2021; Zheng et al., 2022), and more. Many of those CO problems can be formulated within a generic framework of Mixed Integer Linear Programs (MILPs), which is a central focus in algorithm development. Traditional MILP solvers recursively apply a divide-and-conquer strategy to decompose a MILP into sub-problems with additional bounds on the variables in a tree-based search, namely the Branch-&-Bound (B&B) (Land & Doig, 1960), until an optimal solution is found. Off-the-shelf solvers of this kind include SCIP (Achterberg, 2009), CPLEX (Cplex, 2009), and Gurobi (Gurobi Optimization, 2021).

In each iteration, the system solves the relaxed linear program (LP) on a selected node (sub-problem) over the search tree and uses the LP solution (if it contains any fractional variable) to further divide the current problem into two sub-problems. Such traditional solvers heavily rely on hand-craft domain-specific heuristics for branching, which limits their true success and the capability to generalize across domains.

The recent machine learning research has offered new ways to solve MILPs by replacing the need for hand-crafted heuristics with automatically learned heuristics from training data (Gasse et al., 2019; Nair et al., 2020b; Scavuzzo et al., 2022; Parsonson et al., 2023). As a representative example, (Gasse et al., 2019) formulated the MILP with a bipartite graph with variable nodes on the left and constraint nodes on the right, and trained a Graph Neural Network (GNN) to predict the promising variables for branching in B&B. Followup works include the improvements of the GNN models for scaling up (Nair et al., 2018; Gupta et al., 2020) and enhanced solutions (Zarpellon et al., 2021; Qu et al., 2022; Huang et al., 2023b). All of these models are trained via imitation learning (IL), and they thus have one limitation in common, i.e., the effectiveness relies on the availability of highly-quality training data that demonstrates near-optimal branching strategies such as the *full strong branching*

strategy (Achterberg et al., 2005b). Obtaining such training data can be difficult or highly costly in practice, especially for very large graphs where state-of-the-art (SOTA) MILP solvers cannot scale up to produce high-quality demonstrations. Addressing the obstacle, some reinforcement learning (RL) methods have been proposed (Sun et al., 2020; Scavuzzo et al., 2022; Parsonson et al., 2023), which support the learning from scratch without any demonstrations. Nonetheless, those RL-based methods rely on time-consuming online interactions with the solver, which can only be trained over easy MILPs solved in minutes, and have bad transfer performance ( i.e., to be trained on small graphs and tested on large graphs) on evaluation benchmarks (Scavuzzo et al., 2022).

This paper introduces a novel offline RL approach, namely *Ranking-Constrained Actor-Critic (RCAC)*, to address the aforementioned limitations in learning to branch. Different from the standard RL models which rely on online interactions with the environment for collecting training signals, offline RL is directly trained over a *static* data set, which is pre-collected with a certain behavior policy from the environment. Nonetheless, similar to online RL, offline RL harnesses reward information to train the model rather than merely duplicating the training-set behavior, which is the case with imitation learning (IL). Consequently, offline RL can inherit the exploration capability from RL-based MILP solvers on the one hand, and can also significantly reduce the computational cost in training data generation on the other hand. As far as we know, RCAC is the first attempt to apply the offline RL algorithms to MILP solving.

Our empirical results demonstrate the applicability of RCAC to various types of MILPs in the settings of both *exact solving* (without time constraints) and *time-constrained solving*. RCAC consistently outperforms the representative baseline methods across 6 benchmark datasets in terms of both branching quality and training efficiency, including those with hand-crafted heuristics, the IL-based methods, and previous RL-based methods. We present evidence that RCAC behaves better when trained on either sub-optimal datasets containing sparse good demonstrations or small near-optimal datasets collected within a short time. In short, our findings suggest that RCAC holds promise as a potent neural MILP solver for practical applications.

## 2 BACKGROUND

### 2.1 THE B&B ALGORITHM

Each Mixed Integer Linear Program (MILP) is defined by a linear object, linear constraints, and integrality constraints, which can be formally expressed as

$$\min \mathbf{c}^\top \mathbf{x}, \text{ s.t. } \mathbf{A}\mathbf{x} \leq \mathbf{b}, \mathbf{x} \in \mathbb{Z}^p \times \mathbb{R}^{n-p}, \tag{1}$$

where $\mathbf{c} \in \mathbb{R}^n$ represents the objective coefficient vector, $\mathbf{A} \in \mathbb{R}^{m \times n}$ the constraint coefficient matrix, $\mathbf{b} \in \mathbb{R}^m$ the constraint right-hand-side, and $p \leq n$ the number of integer variables. When the integrality constraints are disregarded, we can obtain a linear program (LP) and solve it efficiently with algorithms like the Simplex algorithm. This process is known as linear programming relaxation which will give a lower bound for the original problem since it is solved on a larger feasible region. If the LP relaxed solution $\mathbf{x}^{LP}$ happens to be integral, then $\mathbf{x}^{LP}$ is also guaranteed to be the optimal solution for the original MILP and we are done with the solving. Otherwise, there must be a set of variables $\mathcal{C}$ such that $\mathbf{x}^{LP}[i]$ is fractional for $i \in \mathcal{C}$. The B&B algorithm then selects a variable from $\mathcal{C}$ to partition the problem into two child problems, with the additional constraint

$$\mathbf{x}[i] \leq \lfloor \mathbf{x}^{LP}[i] \rfloor \quad \text{or} \quad \mathbf{x}[i] \geq \lceil \mathbf{x}^{LP}[i] \rceil. \tag{2}$$

This partition process is known as variable selection or branching. With multiple subproblems in hand, each time B&B algorithm will select a subproblem to explore. B&B tracks two pivotal values throughout the solving, the global primal bound (lowest objective value for all feasible solutions) and dual bound (highest objective value for all relaxed solutions), and it continues iterating through the aforementioned steps until the primal bound converges with the dual bound.

The quality of the branching policy has a high impact on the computational cost of B&B. The branching policy needs to balance the size of the search tree and the computational cost for obtaining the branching decision. Among the current heuristics, *full strong branching (FSB)* computes the actual change in the dual bound by solving the resultant subproblem for each fractional variable, which usually achieves the smaller search tree than competing methods (Achterberg et al., 2005a).

However, the computational cost for obtaining the actual bound change itself is expensive. Instead, *pseudocost branching (PB)* (Achterberg et al., 2005a) conducts a fast estimation of the change in bound by averaging the previous changes after branching on each variable, which is faster to compute at the cost of a larger search tree (Achterberg et al., 2005a). In modern solvers, a hybrid branching strategy known as *reliablility pseudocost branching (RPB)* is adopted, which uses FSB at the start of B&B and switches to PB for the remaining steps (Achterberg et al., 2005a).

## 2.2 Reinforcement Learning Formulation for B&B

In standard reinforcement learning (RL), an agent continually interacts with the environment, typically modeled as a Markov Decision Process (MDP). An MDP is defined by a tuple $(\mathcal{S}, \mathcal{A}, p, r, \rho_0, \gamma)$, where $\mathcal{S}$ and $\mathcal{A}$ represent state and action spaces, $p(\mathbf{s}'|\mathbf{s}, a) : \mathcal{S} \times \mathcal{A} \times \mathcal{S} \to [0, 1]$ and $r(\mathbf{s}, a) : \mathcal{S} \times \mathcal{A} \to \mathbb{R}$ represent the state transition and reward functions, $\rho_0(\mathbf{s})$ denotes the initial state distribution and $\gamma \in [0, 1)$ is the discount factor. RL aims to find a policy $\pi(a|\mathbf{s}) : \mathcal{S} \to \mathcal{A}$ that maximizes the expected cumulative discounted rewards, also known as the expected return, denoted as $J(\pi) = \mathbb{E}_{\mathbf{s}_0 \sim \rho_0(\cdot), a_t \sim \pi(\cdot|\mathbf{s}_t), \mathbf{s}_{t+1} \sim p(\cdot|\mathbf{s}_t, a_t)}[\sum_{t=0}^{\infty} \gamma^t r(\mathbf{s}_t, a_t)]$. For each policy $\pi$, it has a corresponding value function $Q^\pi(\mathbf{s}, a)$ which quantifies the expected return when following the policy $\pi$ after taking action $a$ at the state $\mathbf{s}$,

$$Q^\pi(\mathbf{s}, a) = \mathbb{E}_{a_t \sim \pi(\cdot|\mathbf{s}_t), \mathbf{s}_{t+1} \sim p(\cdot|\mathbf{s}_t, a_t)}[\sum_{t=0}^{\infty} \gamma^t r(\mathbf{s}_t, a_t)|\mathbf{s}_0 = \mathbf{s}, a_0 = a]. \quad (3)$$

Assume the reward is bounded, i.e., $|r(\mathbf{s}, a)| \leq R_{max}$, the value function $Q^\pi$ could be computed by iteratively applying the Bellman operator $\mathcal{T}^\pi Q(\mathbf{s}, a) = r(\mathbf{s}, a) + \mathbb{E}_{\mathbf{s}' \sim p(\cdot|\mathbf{s}, a), a' \sim \pi(\cdot|\mathbf{s}')}[Q(\mathbf{s}', a')]$. When $\gamma \in [0, 1)$, the Bellman operator is a contraction (Bertsekas & Tsitsiklis, 1996) with the unique fixed point $Q^\pi(\mathbf{s}, a)$. For the standard Actor-Critic algorithms with the parameterized policy $\pi_\phi$ (actor) and Q-network $Q_\theta$, the update is conducted alternatively between the policy evaluation (Equation 4) and policy improvement (Equation 5),

$$\theta \leftarrow \arg\min_\theta \mathbb{E}_{(\mathbf{s}, a, \mathbf{s}')}[(r(\mathbf{s}, a) + \gamma\mathbb{E}_{a' \sim \pi_\phi(\cdot|\mathbf{s}')}[Q_{\theta'}(\mathbf{s}', a')] - Q_\theta(\mathbf{s}, a))^2], \quad (4)$$

$$\phi \leftarrow \arg\max_\phi \mathbb{E}_{\mathbf{s}}\mathbb{E}_{a \sim \pi_\phi(\cdot|\mathbf{s})}[Q_\theta(\mathbf{s}, a)], \quad (5)$$

where $Q_{\theta'}$ is a slowly updated target Q-function used for a stable estimation of the target Q-value.

The branching inside B&B could also be formulated as an MDP process, with the brancher being the agent and the solver being the environment. Starting from the root node $\mathbf{s}_0$, each time the brancher receives the current B&B search tree as the state $\mathbf{s}$ and selects a variable $a$ from the set of all fractional variables $\mathcal{A}(\mathbf{s})$ for branching. It then receives a manually defined reward $r(\mathbf{s}, a)$, and the solver will partition the problem accordingly to update the search tree to the next state $\mathbf{s}'$. By choosing a reasonable reward function, we can apply RL algorithms to automatically learn a branching policy maximizing the expected return.

## 2.3 Offline Reinforcement Learning

In contrast to standard RL operating in an online setting, offline RL dispenses with real-time interaction. Instead, it trains a policy using a pre-collected dataset $\mathcal{D} = (\mathbf{s}, a, \mathbf{s}', r(\mathbf{s}, a))$. The policy responsible for generating this dataset is referred to as the behavior policy $\pi_\beta(a|\mathbf{s})$. Behavior cloning (BC), one type of imitation learning (IL) method, simply estimates the conditional action distribution from the samples in $\mathcal{D}$ via supervised learning. The performance of BC is highly dependent on the quality of the behavior policy and it typically assumes that the behavior policy is close enough to the optimal policy $\arg\min_\pi J(\pi)$.

In B&B, FSB is usually chosen as the behavior policy for training data generation. Although FSB generally achieves high-quality branching, it could still become sub-optimal when the linear programming relaxation is uninformative or there exists dual degeneracy (Gamrath et al., 2020). Moreover, it is time-consuming to obtain the demonstrations from FSB when it comes to large and hard MILP instances. In comparison, offline RL can make use of the reward information to evaluate the action value like standard RL does, making it a better choice when $\pi_\beta(a|\mathbf{s})$ is sub-optimal or $\mathcal{D}$ is noisy (Kumar et al., 2022).

However, applying online RL algorithms directly to offline RL is also challenging due to the distributional shift between $\pi_\phi$ and $\pi_\beta$ (Kumar et al., 2019; Wu et al., 2019; Jaques et al., 2019; Levine et al., 2020). The Bellman operator in Equation 4 relies on the actions $a'$ sampled from $\pi_\phi(\cdot|\mathbf{s}')$ to estimate target Q-values. When $a'$ falls outside the distribution of actions in dataset $\mathcal{D}$, its Q-value estimation could be arbitrarily wrong. Consequently, $\pi_\phi$ may be biased towards those out-of-distribution (OOD) actions with an erroneously high value when it is optimized to maximize the expected Q-values in Equation 5. Such an error could be corrected via the attempt in the online setting but it can only be avoided in offline RL by constraining the policy $\pi_\phi$ from querying OOD actions. Offline RL algorithms are featured in the implementation of this constraint and we will introduce more details about our solution in the next section.

## 3 METHOD

### 3.1 REWARD FUNCTION FOR BRANCHING

There are multiple ways to measure the quality of the branching strategy such as the solving time, the size of the B&B search tree, the number of iterations spent in solving LP, and dual integrals. We finally choose the improvement of the dual bound, $|\mathbf{c}^\top \mathbf{x}_{t+1}^{LP} - \mathbf{c}^\top \mathbf{x}_t^{LP}|$, as the reward function due to the following reasons. First, different from the metrics involving time measurements like per-step solving time and dual integrals, its value is not dependent on the time cost for obtaining the branching decision and is invariant to operating systems. Second, the dual-bound improvement can somewhat serve as a direct indicator of the quality of the branching decision that led to its value, whereas metrics such as the number of LP iterations and the change in the search tree's size do not provide this insight. Finally, the cumulative discounted value of the dual bound improvement is still informative when an MILP instance is not solved exactly but stopped at a given time limit, compared with the cumulative discounted number of LP iterations or increased size of the search tree. The discount factor $\gamma$ also favors an early improvement of the dual bound in the expected return as the dual integral does.

### 3.2 RANKING-CONSTRAINED ACTOR-CRITIC ALGORITHM

Since the Q-value estimated at a rarely explored action is imprecise in offline RL, one intuitive approach is to restrict the policy $\pi_\phi$ from selecting actions that have a low probability density in the dataset (Wu et al., 2019; Fujimoto et al., 2019; Kumar et al., 2019). Nevertheless, when the behavior policy is sub-optimal, those high-quality actions almost surely have a low probability density and will be unavoidably excluded by such a strict constraint. In fact, a good action does no harm to policy optimization even if it is an OOD action. Therefore, our idea is to balance the quality and probability density of actions when we try to filter out those toxic OOD actions.

Normally, there is no way to tell if an action is good or not in offline RL until we have evaluated its Q-value. While in B&B, we can use the dual-bound improvement it brings, which is also the reward function we use, to coarsely evaluate its branching quality as FSB does. So we first train a scoring function $G_\omega(a|\mathbf{s})$ which uses different weights to maximize the log-likelihood of an action in the dataset given the reward it obtains. The training objective could be expressed as

$$\arg\min_\omega \mathbb{E}_{(\mathbf{s},a,r(\mathbf{s},a))\sim\mathcal{D}}[-(\lambda\mathbb{1}_{r(\mathbf{s},a)>\zeta} + 1)\log G_\omega(a|\mathbf{s})], \tag{6}$$

where $\lambda \geq 0$ is a factor promoting the actions leading to a dual-bound improvement greater than $\zeta$. In most cases, $\zeta$ can be simply set as zero due to the sparse reward nature of the environment.

We then constrain the policy $\pi_\phi$ by depressing the Q-value for actions out of the top $k$ candidates at state $\mathbf{s}$ ranked by $G(a|\mathbf{s})$ using a large negative value $-\delta$, i.e.,

$$\bar{Q}(\mathbf{s},a) = \begin{cases} Q(\mathbf{s},a), & \text{if } a \in \text{top } k(G_\omega(a|\mathbf{s})), \\ -\delta, & \text{otherwise.} \end{cases} \tag{7}$$

We then refine the policy evaluation and policy improvement in the Actor-Critic algorithm as

$$\theta \leftarrow \arg\min_\theta \mathbb{E}_{(\mathbf{s},a,\mathbf{s}')\sim\mathcal{D}}[(r(\mathbf{s},a) + \gamma\mathbb{E}_{a'\sim\pi_\phi(\cdot|\mathbf{s}')}[\bar{Q}_{\theta'}(\mathbf{s}',a')] - Q_\theta(\mathbf{s},a))^2], \tag{8}$$

$$\phi \leftarrow \arg\max_\phi \mathbb{E}_{\mathbf{s}\sim\mathcal{D},a\sim\pi_\phi(\cdot|\mathbf{s})}[\bar{Q}_\theta(\mathbf{s},a)]. \tag{9}$$

We refer to our method as *Ranking-Constrained Actor-Critic (RCAC)* algorithm. The ranking constraint could alternatively realized by the relative ranking as top $k\%$ candidates, but using an absolute rank is more friendly to the batch operation in neural network training. During inference, the action $\arg\max_a \pi_\phi(a|\mathbf{s})$ is used at each step. Our algorithm could be summarized as follows

---

**Algorithm 1** Ranking-Constrained Actor-Critic

---

1: **Input**: Dataset $\mathcal{D} = \{(\mathbf{s}, a, \mathbf{s}', r(\mathbf{s}, a))\}$,
2: Randomly initialize ranking model $G_\omega$, policy network $\pi_\phi$ and Q-network $Q_\theta$
3: Pretrain $G_\omega$ with the loss defined in Equation 6
4: **for** iteration $i = \{1, \cdots, I\}$ **do**
5:     Sample a batch of transitions $\mathcal{B}$ from $\mathcal{D}$
6:     $\theta \leftarrow \arg\min_\theta \mathbb{E}_{(\mathbf{s},a,\mathbf{s}',r(\mathbf{s},a))\sim\mathcal{B}}[(r(\mathbf{s}, a) + \gamma\mathbb{E}_{a'\sim\pi_\phi(\cdot|\mathbf{s}')}[\bar{Q}_{\theta'}(\mathbf{s}', a')] - Q_\theta(\mathbf{s}, a))^2]$
7:     $\phi \leftarrow \arg\max_\phi \mathbb{E}_{\mathbf{s}\sim\mathcal{B}, \mathbf{a}\sim\pi_\phi(\cdot|\mathbf{s})}[\bar{Q}_\theta(\mathbf{s}, a)]$
8:     $\theta' \leftarrow \tau\theta' + (1-\tau)\theta$
9: **end for**
10: **return** $\pi_\phi$

---

### 3.3 Modeling the B&B Tree

We use a bipartite graph representation for the B&B node, where $\mathcal{G} = (\mathbf{V}, \mathbf{C}, \mathbf{E})$, with variable node features $\mathbf{V} \in \mathbb{R}^{n \times d_1}$, constraint node features $\mathbf{C} \in \mathbb{R}^{m \times d_3}$ and edge features $\mathbf{E} \in \mathbb{R}^{n \times m \times d_2}$. We use the same features and GNN architecture from Gasse et al. (2019), where the model architecture is kept the same for $G_\omega$, $\pi_\phi$ and $Q_\theta$. We normalize both node and edge features in the dataset. For example, given the $i$-th feature of a node $j$, we will normalize it as $\mathbf{V}[j, i] \leftarrow (\mathbf{V}[j, i] - \mu_i^v)/(\sigma_i^v)$, where $\mu_i^v$ and $\sigma_i^v$ are the estimated mean and standard deviation for the $i$-th dimension of node features. The constraint features and edge features are similarly processed.

## 4 Experiments

### 4.1 Experimental Setup

**Baselines.** We compare our method to two classical branching heuristics, full strong branching (FSB) and reliability pseudocost branching (PRB), and two neural methods, including the online RL method tree MDP (tMDP) (Scavuzzo et al., 2022), and IL method GGCN (Gasse et al., 2019). Besides, we also design a vanilla hybrid branching (VHB) heuristic, which adopts FSB with probability 0.05 at each decision step and uses the pseudocost branching otherwise. VHB will serve as one type of behavior policy for our demonstration collection.

**Metrics.** We use two different types of metrics in previous evaluations (Gasse et al., 2019; Nair et al., 2018; Gasse et al., 2022) for B&B methods. The first type of metric evaluates the model's efficiency for exact solving without any time constraint, including the total solving time and the size of the B&B search tree (measured by its number of nodes). The former is a universal metric to compare both neural methods and hand-crafted heuristics, while the latter is more straightforward for comparison when the decision time is basically the same, as in the neural methods. The second type of metric evaluates the quality of the dual bound when solving is constrained by a given time limit $T$, here we use the dual-integral metric, $T\mathbf{c}^\top\mathbf{x}^* - \int_{t=0}^T z_t^* dt$, where $\mathbf{c}^\top\mathbf{x}^*$ is the optimal objective value and $z_t^*$ is the best dual bound at time $t$. $T$ is set as 15 minutes in our experiment.

**Benchmarks.** We evaluate our method on six commonly used MILP benchmarks, including four synthetic easy problems and two hard problems from real-world applications, as listed in Table 1. They are categorized as easy and hard problems according to the time for exact solving, where MILP instances from easy problems can all be solved by SCIP (version 7.0.3) within 10 minutes, and MILP instances from hard problems take SCIP more than 1 hour to finish on average. The four synthetic easy problems include *Set Covering (SC)*, *Maximal Independent Set (MIS)*, *Combinatorial Auction (CA)* and *Capacitated Facility Location (CFL)*. We follow the same instance generation process as in Gasse et al. (2019) to generate 10,000 MILP instances for training, 2,000 instances for validation,

and 20 instances for testing on each problem. We use the solving time and search tree size as the evaluation metric since their instances can all be solved in a short time. The two hard problems, *Workload Apportionment* (WA) and *Anonymous Problem (AP)*, are from the ML4CO competition (Gasse et al., 2022). We use their existing training, validation, and testing split. In light of the difficulty in solving their instances exactly, we use the dual integral as the evaluation metrics for these two problems. Since obtaining the optimal solution $\mathbf{x}^*$ is hard in practice and it does not affect the comparison among methods, we directly report the score from the ML4CO evaluation script, which is a negated unshifted version of the dual integral intended to be maximized. Additional details about the instances for each benchmark are available in the Appendix.

To generate the demonstrations for training, we consider two different scenarios. In the first scenario, we assume we only have access to a sub-optimal heuristic on a certain problem. We *simulate* this heuristic with VHB and use it to generate a dataset with 100,000 transitions. In the second scenario, we still have access to the near-optimal heuristic, but due to its expensive cost, we can only generate a small dataset for training. We use FSB, which has been empirically shown to be near-optimal on the benchmarks we consider, to generate 5,000 transitions on each problem, whose size is only 5% of the standard dataset size used for training previous IL methods (Gasse et al., 2019; Gupta et al., 2020). We compare the generation time for both our datasets and standard datasets in Table 1.

| Dataset Prefix | Problem | Time for Our VHB Dataset | Time for Our FSB Dataset | Time for Standard FSB Dataset |
|---|---|---|---|---|
| SC | Set Covering | 0.3 h | 0.2 h | 1.0 h |
| MIS | Maximum Independent Set | 1.1h | 0.4 h | 4.5 h |
| CA | Combinatorial Auction | 0.2 h | 0.1 h | 0.8 h |
| CFL | Capacitated Facility Location | 2.2 h | 1.0 h | 7.2 h |
| WA | Workload Apportionment | 21.2 h | 13.3 h | 266.4 h |
| AP | Anonymous Problem | 1.1 h | 1.1 h | 6.4 h |

Table 1: Dataset Collection Statistics. We employ 20 parallel SCIP solvers to collect demonstrations for each dataset. The collection time is in hours. Results show that collecting demonstrations for the standard FSB dataset is much more expensive than our VHB dataset and small FSB dataset.

## 4.2 Efficiency for Exact Solving

We first evaluate RCAC in its efficiency for the exact solving of MILPs. We train RCAC and GGCN on both the sub-optimal dataset collected by VHB (denoted with 'H') and a small near-optimal dataset collected by FSB (denoted with 'S'). Five random seeds are used during the training and testing for each method. We compare the solving time and the size of the search tree in Figure 1, and report the mean and standard deviation in Table 2 and 3.

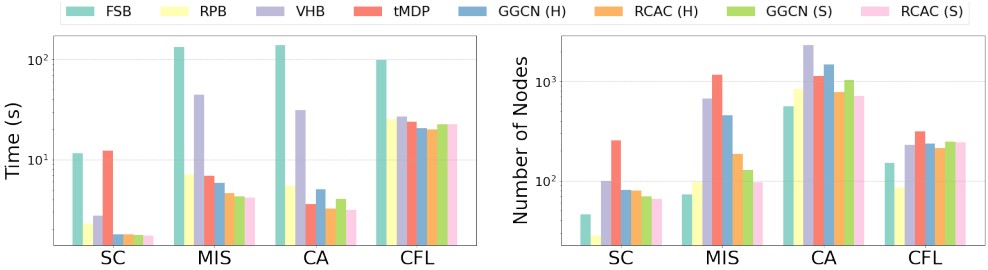

Figure 1: Comparison among all methods in the solving time (left) and size of the search tree (right) on SC, MIS, CA, and CFL. The y-axis is in the log scale.

Compared with non-neural baselines, both RCAC and GGCN have shown clear advantages in solving MILPs exactly with less time, though trained on a sub-optimal dataset or a smaller near-optimal dataset. Besides, it can be clearly observed that RCAC is better than GGCN across all benchmarks and two types of training datasets in both solving time and the number of nodes, especially on MIS and CA. Different from GGCN which simply learns the conditional action distribution from

| Model | SC Time (s) ↓ | MIS Time (s) ↓ | CA Time(s) ↓ | CFL Time(s) ↓ |
|---|---|---|---|---|
| FSB | $11.61 \pm 0.28$ | $134.38 \pm 5.50$ | $140.26 \pm 2.55$ | $98.65 \pm 9.87$ |
| RPB | $2.31 \pm 0.04$ | $7.10 \pm 0.17$ | $5.51 \pm 0.04$ | $25.23 \pm 0.98$ |
| VHB | $2.76 \pm 0.14$ | $44.37 \pm 6.64$ | $31.45 \pm 1.56$ | $26.85 \pm 1.91$ |
| tMDP | $12.24 \pm 0.05$ | $6.92 \pm 2.94$ | $3.56 \pm 0.10$ | $24.06 \pm 0.41$ |
| GGCN (H) | $1.78 \pm 0.05$ | $5.86 \pm 0.31$ | $5.06 \pm 0.23$ | $20.71 \pm 1.66$ |
| RCAC (H) | $1.78 \pm 0.04$ | $4.64 \pm 0.19$ | $3.22 \pm 0.09$ | **$19.94 \pm 0.50$** |
| GGCN (S) | $1.76 \pm 0.07$ | $4.29 \pm 0.13$ | $4.05 \pm 0.11$ | $22.63 \pm 0.96$ |
| RCAC (S) | **$1.73 \pm 0.04$** | **$4.13 \pm 0.14$** | **$3.15 \pm 0.06$** | $22.47 \pm 1.31$ |

Table 2: Comparative results in time for exact solving on SC, MIS, CA and CFL. We bold the best results and color the second-best results in green on each dataset.

| Model | SC # Nodes ↓ | MIS # Nodes ↓ | CA # Nodes ↓ | CFL # Nodes ↓ |
|---|---|---|---|---|
| FSB | $46.0 \pm 0.1$ | $73.0 \pm 4.5$ | $559.7 \pm 7.51$ | $150.7 \pm 4.7$ |
| RPB | $28.0 \pm 2.4$ | $96.7 \pm 14.3$ | $840.2 \pm 49.6$ | $86.1 \pm 12.2$ |
| VHB | $99.1 \pm 6.8$ | $677.0 \pm 141.4$ | $2330.3 \pm 115.9$ | $228.4 \pm 12.1$ |
| tMDP | $254.9 \pm 20.9$ | $1163.1 \pm 1295.9$ | $1136.2 \pm 55.1$ | $316.1 \pm 30.4$ |
| GGCN (H) | $80.8 \pm 7.3$ | $454.5 \pm 77.8$ | $1471.6 \pm 94.5$ | $235.2 \pm 14.9$ |
| RCAC (H) | $80.3 \pm 13.7$ | $185.2 \pm 34.8$ | $774.9 \pm 27.8$ | **$211.9 \pm 14.2$** |
| GGCN (S) | $69.3 \pm 8.2$ | $127.7 \pm 29.9$ | $1040.0 \pm 36.0$ | $246.5 \pm 22.1$ |
| RCAC (S) | **$65.9 \pm 6.1$** | **$96.4 \pm 16.8$** | **$718.8 \pm 24.5$** | $242.3 \pm 15.7$ |

Table 3: Comparative results in the size of search tree for exact solving on SC, MIS, CA and CFL. Human heuristics and neural methods are above and under the line separately. We bold the best results and color the second-best results for *neural methods* in green on each dataset.

the dataset, RCAC can utilize the reward information to evaluate the quality of actions. We want to highlight that such a capability can also explain the success of RCAC on smaller near-optimal datasets. Typically, a high-quality branching decision in the first few steps of B&B will have a more profound impact on the size of the search tree, as the spirit of RPB suggests. Since the dual-bound improvement is also larger at the earlier stage, Equation 5 will then encourage RCAC to place more emphasis on learning good actions in the first few steps of B&B due to a large Q-value at this time. While GGCN just equally imitates the branching decisions at all periods of B&B, resulting in an inferior performance than RCAC when the data is in short. Finally, although tMDP could sometimes achieve a good performance such as on CA, its overall performance is still worse than GGCN and RCAC trained on a sub-optimal or a small near-optimal dataset, not to mention its overwhelming training time which could amount to six days. In comparison, the data collection and training of RCAC only takes a few hours and it achieves a much better branching performance. Therefore, though based on the same motivation to overcome the limitations in collecting datasets with FSB, training RCAC from sub-optimal or small near-optimal datasets is clearly better than training an RL agent from scratch. All these findings combine to justify RCAC's advantage over both IL and RL methods in learning to branch for exact solving.

## 4.3 DUAL INTEGRAL FOR TIME-CONSTRAINED SOLVING

We then evaluate RCAC on two hard problems, WA and AP, in the dual-integral score. We exclude tMDP on these two datasets due to its long training time and bad performance on easy problems. We evaluate each model on 20 testing instances from the official split and report the best results for each model. We compare the model performance in Figure 2 and Table 4.

Basically, neural methods do not show a very strong advantage against non-neural methods possibly due to the hardness of the problems themselves. But we can still see that RCAC shows some promising signals. When trained on a small near-optimal dataset, RCAC takes the lead of all methods on WA and is the second-best one on AP. Besides, RCAC outperforms GGCN when trained on both types of datasets, especially on AP where dense rewards exist in the environment. This evidence

indicates RCAC's potential to improve training efficiency on hard problems like WA, where the data collection time could amount to days or weeks.

| Model | WA Score ↑ | AP Score ↑ |
|---|---|---|
| FSB | 633653 | 25411832 |
| RPB | 634846 | **27368259** |
| VHB | 633837 | 25411230 |
| tMDP | - | - |
| GGCN (H) | 635072 | 25308238 |
| RCAC (H) | 635099 | 25311504 |
| GGCN (S) | 635074 | 25430097 |
| RCAC (S) | **635103** | 25564703 |

Table 4: Comparative results in the score (negated and shifted version of dual-integral, to be maximized) on WA and AP. We bold the best results and color the second-best results in green on each dataset.

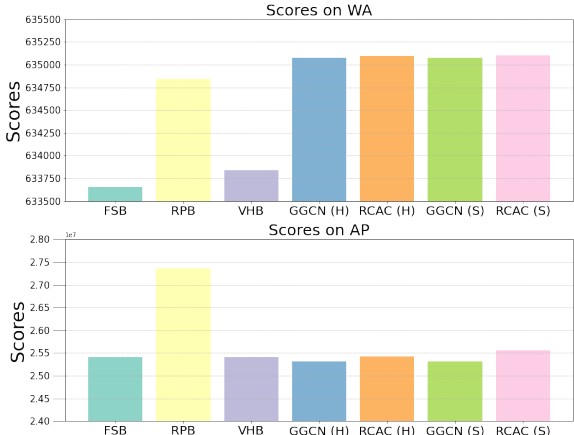

Figure 2: Comparison among all methods except tMDP on WA and AP datasets.

### 4.4 ABLATION STUDY

Although our ranking model relies on dual-bound information to roughly evaluate the quality of a branching decision similar to FSB, RCAC is still different from imitating FSB since it maximizes not the instant dual-bound change but the long-term cumulative rewards. To further understand the source of improvement from RCAC, we use the models trained on the hybrid branching dataset as an example for ablation. We compare the testing performance of the pretrained $G_\omega$ used for the final training of RCAC in Table 5. Basically, it is undeniable that the strong performance of $G_\omega$ largely

| | SC # Nodes ↓ | MIS # Nodes ↓ | CA # Nodes ↓ | CFL # Nodes ↓ |
|---|---|---|---|---|
| $G_\omega$ (H) | **76.9 ± 2.3** | 213.4 ± 34.2 | 886.8 ± 58.0 | 217.1 ± 14.9 |
| RCAC (H) | 80.3 ± 13.7 | **185.2 ± 34.8** | **774.9 ± 27.8** | **211.9 ± 14.2** |

Table 5: Comparison between the performance of the ranking model $G_\omega$ and RCAC trained on the sub-optimal dataset collected by vanilla hybrid branching.

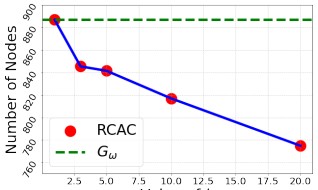

Figure 3: Effect of $k$ in RCAC's performance on CA.

contributes to the improvement in RCAC, where $G_\omega$ has already shown a comprehensive advantage against GGCN trained on the same dataset. But we can also observe that in most cases, RCAC can further improve the performance of $G_\omega$, which is most prominent on CA. To further understand the exploration ability of RCAC, we visualize the effect of $k$ on RCAC's performance on the CA dataset in Figure 3. It can be seen that as $k$ increases, the number of nodes keeps decreasing. This suggests that RCAC is not simply doing knowledge distillation (Gupta et al., 2020) from $G_\omega$ but learning to evaluate the Q-value for the top candidates ranked by $G_\omega$ and maximize the expected return.

## 5 RELATED WORK

### 5.1 NEURAL MILP SOLVERS

Traditional MILP solvers rely on plenty of hand-crafted heuristics during their execution. Neural solvers thus aim to improve these heuristics with deep learning methods (Bengio et al., 2021). Current neural solvers have successfully improved the performance of neural solvers by learning the heuristics in variable selection (branching) (Gasse et al., 2019; Gupta et al., 2020; Nair et al., 2020b; Zarpellon et al., 2021; Scavuzzo et al., 2022; Huang et al., 2023b), node selection (He et al., 2014; Song et al., 2018), cutting plane selection (Tang et al., 2020; Paulus et al., 2022; Turner et al.,

2023; Wang et al., 2023), large neighborhood search (Sun et al., 2020; Wu et al., 2021; Sonnerat et al., 2021; Huang et al., 2023a), diving (Nair et al., 2020b; Yoon, 2022; Han et al., 2023; Paulus & Krause, 2023) and primal heuristics selection (Khalil et al., 2017; Hendel et al., 2019; Chmiela et al., 2021). Our work studies the variable selection heuristic, which receives the most attention in neural solvers.

(Khalil et al., 2016; Alvarez et al., 2017; Hansknecht et al., 2018) are the earliest works to use statistical learning for the branching heuristic. They use an imitation learning method to first collect an offline dataset with full strong branching and then treat the learning as either a ranking (Khalil et al., 2016; Hansknecht et al., 2018) or regression problem (Alvarez et al., 2017). With the advent of GNNs, (Gasse et al., 2019) transform each MILP instance into a bipartite graph consisting of variable nodes and constraint nodes and train a GNN classifier to imitate the choice of strong branching. This work lays out the basic model architecture for neural solvers on variable selection. To extend this GNN-based neural solver to larger instances, (Nair et al., 2020b) adopt a more efficient batch Linear Programming solver based on the alternating direction method of multipliers. Furthermore, (Gupta et al., 2020) improves the low efficiency of GNNs by using a hybrid model. In detail, they extract the structural information for each MILP with GNN once at the root node and then use a fast multi-layer perceptron to do the classification at each node with the extracted structural information and current node features. Recently, (Scavuzzo et al., 2022) proposed a reinforcement learning approach for learning to branch by formulating B&B as a tree-structured MDP process while Parsonson et al. (2023) utilize RL to efficiently learn from retrospective trajectories. (Huang et al., 2023b) and (Qu et al., 2022) are the two most similar methods to our work. Although both of the methods are claimed to be offline RL methods, they are actually different from the offline RL algorithms featured in dealing with OOD actions. Namely, they still assume cheap access to a near-optimal expert heuristic without considering a sub-optimal dataset. Therefore, our method is de facto the first work in applying offline RL in learning to branch.

### 5.2 Offline Reinforcement Learning

Offline RL has wide applications in robotic manipulation (Kalashnikov et al., 2018; Mandlekar et al., 2019; Singh et al., 2021; Kalashnikov et al., 2021), text generation (Jaques et al., 2020; Snell et al., 2023), and healthcare (Shortreed et al., 2010; Wang et al., 2018), but it is known to suffer from the distributional shift problem (Kumar et al., 2019; Wu et al., 2019; Jaques et al., 2019; Levine et al., 2020). Existing methods generally tackle this challenge by restricting the policy from generating the OOD actions via an explicit density model (Wu et al., 2019; Fujimoto et al., 2019; Kumar et al., 2019; Ghasemipour et al., 2020), implicit divergence constraint (Peng et al., 2019; Nair et al., 2020a; Wang et al., 2020; Kostrikov et al., 2022; Li et al., 2023), conservative estimation of state-action value (Kumar et al., 2020; Kostrikov et al., 2021; Lyu et al., 2022), or adding a behavior cloning term to the policy improvement objective (Nair et al., 2018; Fujimoto & Gu, 2021). Our model is mostly relevant to the offline RL methods in the first category, but we further tackle the challenge from the dynamic action space and incorporate the unique information from the B&B algorithm.

Compared with imitation learning methods such as behavior cloning, offline RL can be more robust to noisy or suboptimal demonstrations (Kumar et al., 2022). Therefore, our proposed offline RL method no longer relies on a near-optimal expert policy as previous neural solvers do and becomes more flexible in the data collection process.

## 6 Conclusion

In this paper, we propose a novel offline RL approach RCAC for neural branching in mixed linear integer programming. RCAC tackles the limitations of previous neural branching algorithms in their dependence on near-optimal human heuristics and the high cost of data collection. It outperforms previous IL-based and RL-based neural branching methods in both branching quality and training efficiency, for both exact solving and time-constrained solving. RCAC thus exhibits a strong potential in generalizing neural MILP solvers to more challenging problems. Future extension of this work includes (1) the combination of both online and offline RL training, (2) the consideration of the multitasking nature of MILP solving, and (3) the generalization of RCAC to other heuristics in MILP solving such as diving and large neighborhood search.

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

## A    APPENDIX

## B    ADDITIONAL BENCHMARK DETAILS

For four synthetic easy problems, we use the same instance generation methods in (Gasse et al., 2019) to generate all instances using the Ecole library (Prouvost et al., 2020), we include both the generation method and key parameters in Table 6. We also report the average number of variables, integer variables, and constraints across all training and testing instances in the same table. The majority of datasets exhibit consistent sizes with minimal variance across all MILP instances, except for those in the Anonymous Problem. Anonymous Problem is in fact a curated set of the well-known MIPLIB benchmark (Gleixner et al., 2021), which is a heterogeneous dataset consisting of mixed integer programs with various types, structures, and sizes. The Anonymous Problem dataset encompasses MILP instances with variable counts ranging from 1,613 to 92,261, integer variables spanning from 162 to 36,960, and constraints numbering between 1,080 and 126,621.

## C    ADDITIONAL TRAINING DETAILS

To train the IL baseline GGCN (Gasse et al., 2019), we follow the procedure in its official code base [1]. We use the validation instances to generate an additional validation set with 20,000 transitions for the dataset collected by vanilla hybrid branching and 1,000 transitions for the dataset collected by full strong branching. The ratio between the size of the training set and validation set is kept as 5:1. We train GGCN for a maximum of 1,000 epochs, and each epoch contains 312 mini-batches with size 32, which are sampled from the whole dataset with replacement. The model is optimized by an Adam optimizer with an initial learning rate of 0.001. The learning rate is divided by 5 whenever the accuracy on the validation set does not increase in 10 straight epochs. We also adopt the early stopping strategy to terminate the training if the validation accuracy does not improve in 20 straight epochs.

---

[1] `https://github.com/ds4dm/learn2branch`

| Dataset | Generation Method | Parameters | # Variables | # Int. Variables | # Constraints |
|---|---|---|---|---|---|
| Set Covering | Balas & Ho (1980) | Rows: 500 Columns: 1000 | 1,000 | 1000 | 500 |
| Maximum Independent Set | Bergman et al. (2016) Erdos & Rényi (1984) | Nodes: 500 Affinity: 4 | 500 | 500 | 2,087 |
| Combinatorial Auction | Leyton-Brown et al. (2000) | Items: 100 Bids: 500 | 500 | 500 | 193 |
| Capacitated Facility Location | Cornuejols et al. (1991) | Customers: 100 Facilities: 100 | 10,100 | 100 | 10,201 |
| Workload Aportionment | (Gasse et al., 2022) | - | 61,000 | 1,000 | 64,321 |
| Anonymous Problem | (Gasse et al., 2022) | - | 34,656 | 13,533 | 44,430 |

Table 6: Instance generation details and sizes of MILP instances in each dataset. The average number of variables, integer variables (abbreviated as Int. Variables), and constraints are presented.

For tMDP (Scavuzzo et al., 2022), due to its excessive training time which could take up to six days for a single model, we directly use the best checkpoints from its public code repository [2] on four synthetic easy problems. The model checkpoint for each problem is tested on five different random seeds as well. Because tMDP was initially developed for exact solving, and considering its substantial training expenses and poor performance on the four synthetic datasets, we refrain from comparing our RCAC to it on the WA and AP datasets.

For RCAC, since no simple measurement like validation accuracy can be used to evaluate its performance, we hold 20 instances from the validation set to track its solving performance directly. We use the number of nodes as the validation metric on SC, MIS, CA, and CFL and the scores from ML4CO (Gasse et al., 2022) as the validation metric on WA and AP. RCAC's training is two-fold, which involves the pretraining of $G_\omega$ and the training of the actor $\pi_\phi$ and critic $Q_\theta$. For simplicity, we keep the training manner the same in both stages. In each stage, we train the model for a maximum of 100 epochs, each epoch contains 624 mini-batches with size 32. We use an Adam optimizer with an initial learning rate of 0.0001 for each network and divide the learning rate by 5 if no improvement in the validation performance is seen in the last 3 epochs. The training will be terminated earlier if the validation performance does not improve in 5-straight epochs.

For other hyperparameters of RCAC, we detail the selection process as follows:

1. Threshold $\zeta$. We simply set $\zeta$ as the median among all rewards in the dataset. For four synthetic easy problems, $\zeta = 0$ due to the sparse nature of the environment.

2. Factor $\lambda$. We search it from the set of $\{1, 5, 10\}$ and use the one with the highest validation performance.

3. Ranking threshold $k$. This is the most important hyperparameter in RCAC. We search it over $\{3, 5, 10, 20\}$ for the one with the highest validation performance. Although in some cases $k = 1$ achieves the best result, such as on the SC dataset, we do not take this special case corresponding to simply cloning $G_\omega$ into consideration when we evaluate RCAC. For practical use, it is free to choose the one from $G_\omega$ and $\pi_\phi$ with the higher validation performance. Typically, a small $k$ is preferred on a dataset if the demonstrations have a high quality. We include the detailed choice of $k$ for each dataset in Table 7.

4. Negative penalization $-\delta$. We set it as $-10^6$. Basically, any large negative value could serve the purpose and RCAC is insensitive to its choice.

## D   ADDITIONAL EXPERIMENTS

Here we first perform the t-test for the hypothesis that RCAC is better than GGCN on the results from Table 2 and 3. The $p$-values are shown in Table 8.

---

[2] https://github.com/lascavana/rl2branch

|  | SC | MIS | CA | CFL |
|---|---|---|---|---|
| Dataset (H) | 3 | 10 | 20 | 3 |
| Dataset (S) | 10 | 3 | 5 | 3 |

Table 7: Value of the hyperparamaeter $k$ on each dataset for RCAC.

|  | SC | MIS | CA | CFL |
|---|---|---|---|---|
| Time (H) | 0.50 | $2.7 \times 10^{-4}$ | $2.9 \times 10^{-6}$ | 0.43 |
| Time (S) | 0.24 | 0.07 | $23.6 \times 10^{-6}$ | 3 |
| # Nodes (H) | 0.48 | $3.7 \times 10^{-4}$ | $3.7 \times 10^{-6}$ | 0.032 |
| # Nodes (S) | 0.27 | 0.059 | $3.0 \times 10^{-6}$ | 0.38 |

Table 8: $p$-value of the t-test for the hypothesis that RCAC is better than GGCN on the results from Table 2 and 3.

We further test all methods on a larger testing set, with 100 instances and 5 seeds for each problem, and report the performance in Table 9 and 10. It can be seen that RCAC still holds its advantage on this even larger testing set.

|  | SC | MIS | CA | CFL |
|---|---|---|---|---|
| Model | Time (s) ↓ | Time (s) ↓ | Time(s) ↓ | Time(s) ↓ |
| FSB | $6.17 \pm 0.12$ | $111.96 \pm 4.82$ | $84.79 \pm 1.18$ | $105.96 \pm 1.83$ |
| RPB | $1.97 \pm 0.02$ | $6.24 \pm 0.09$ | $4.13 \pm 0.04$ | $30.55 \pm 1.06$ |
| VHB | $2.06 \pm 0.09$ | $30.60 \pm 2.03$ | $19.03 \pm 0.27$ | $38.52 \pm 2.09$ |
| tMDP | $2.01 \pm 0.02$ | $4.61 \pm 0.45$ | $3.00 \pm 0.16$ | $34.39 \pm 1.00$ |
| GGCN (H) | $1.71 \pm 0.02$ | $5.00 \pm 0.26$ | $4.51 \pm 0.21$ | $30.84 \pm 0.50$ |
| CQL (H) | $1.71 \pm 0.03$ | $4.48 \pm 0.09$ | $2.70 \pm 0.17$ | $31.23 \pm 1.49$ |
| RCAC (H) | $1.69 \pm 0.05$ | $4.26 \pm 0.06$ | $2.65 \pm 0.07$ | $\mathbf{30.28 \pm 0.79}$ |
| GGCN (S) | $1.66 \pm 0.01$ | $3.91 \pm 0.97$ | $2.49 \pm 0.02$ | $32.05 \pm 2.06$ |
| C QL (S) | $1.68 \pm 0.04$ | $4.10 \pm 0.27$ | $3.84 \pm 0.36$ | $31.80 \pm 1.69$ |
| RCAC (S) | $\mathbf{1.65 \pm 0.02}$ | $\mathbf{3.86 \pm 0.32}$ | $\mathbf{2.46 \pm 0.04}$ | $31.68 \pm 1.60$ |

Table 9: Comparative results in time for exact solving on SC, MIS, CA and CFL. Models are evaluated on 100 testing instances with 5 seeds for each dataset. We bold the best results and color the second-best results in green on each dataset.

|  | SC | MIS | CA | CFL |
|---|---|---|---|---|
| Model | # Nodes ↓ | # Nodes ↓ | # Nodes ↓ | # Nodes ↓ |
| FSB | $33.3 \pm 0.5$ | $74.92 \pm 3.74$ | $359.1 \pm 5.10$ | $172.6 \pm 8.1$ |
| RPB | $14.1 \pm 0.8$ | $104.1 \pm 6.6$ | $424.5 \pm 25.7$ | $276.4 \pm 5.0$ |
| VHB | $67.81 \pm 1.32$ | $592.0 \pm 43.9$ | $1486.5 \pm 17.6$ | $423.2 \pm 33.1$ |
| tMDP | $171.3 \pm 8.4$ | $724.1 \pm 698.2$ | $746.9 \pm 55.1$ | $632.0 \pm 26.1$ |
| GGCN (H) | $57.1 \pm 1.3$ | $321.0 \pm 53.9$ | $1026.4 \pm 46.8$ | $505.1 \pm 10.2$ |
| CQL (H) | $58.2 \pm 4.0$ | $246.8 \pm 90.9$ | $592.7 \pm 86.6$ | $526.0 \pm 20.0$ |
| RCAC (H) | $53.9 \pm 6.8$ | $161.7 \pm 17.0$ | $503.9 \pm 5.9$ | $\mathbf{487.5 \pm 19.9}$ |
| GGCN (S) | $42.6 \pm 0.4$ | $94.7 \pm 9.6$ | $432.0 \pm 6.5$ | $545.1 \pm 44.7$ |
| CQL (S) | $51.3 \pm 1.4$ | $104.2 \pm 17.2$ | $995.7 \pm 680.8$ | $541.2 \pm 40.3$ |
| RCAC (S) | $\mathbf{41.8 \pm 1.6}$ | $\mathbf{88.7 \pm 10.1}$ | $\mathbf{421.8 \pm 4.0}$ | $531.0 \pm 27.0$ |

Table 10: Comparative results in the size of search tree for exact solving on SC, MIS, CA and CFL. Models are evaluated on 100 testing instances with 5 seeds for each dataset. Human heuristics and neural methods are above and under the line separately. We bold the best results and color the second-best results for *neural methods* in green on each dataset.

Besides, in order to evaluate the generalization performance of RCAC, we directly run the trained models of RCAC on 40 larger instances for each problem. In detail, we generate SC instances

with 1,000 rows and 1,000 columns; MIS instances with 1,000 nodes with affinity 4, CA instances with 200 items and 1,000 bids; and CFL instances with 200 customers and 100 facilities. The model performances are reported in Table 11 and 12. Though RCAC shows inferior performance to GGCN on the SC dataset, it still holds the advantage over the other three datasets when trained on both sub-optimal and small datasets.

| Model | SC Time (s) ↓ | MIS Time (s) ↓ | CA Time(s) ↓ | CFL Time(s) ↓ |
|---|---|---|---|---|
| FSB | $107.50 \pm 1.59$ | $558.60 \pm 100.84$ | $752.01 \pm 65.79$ | $295.12 \pm 5.29$ |
| RPB | $8.82 \pm 0.10$ | $112.20 \pm 1.68$ | $64.96 \pm 1.60$ | $78.13 \pm 1.17$ |
| VHB | $16.32 \pm 0.09$ | $406.60 \pm 29.51$ | $591.97 \pm 23.45$ | $89.67 \pm 3.01$ |
| tMDP | $17.89 \pm 1.09$ | $114.81 \pm 10.47$ | $125.88 \pm 6.24$ | $90.43 \pm 2.20$ |
| GGCN (H) | $7.77 \pm 0.08$ | $118.16 \pm 9.16$ | $96.12 \pm 11.59$ | $74.24 \pm 1.81$ |
| RCAC (H) | $8.08 \pm 0.84$ | $112.77 \pm 33.91$ | $81.34 \pm 1.58$ | $\mathbf{73.23 \pm 0.80}$ |
| GGCN (S) | $\mathbf{6.71 \pm 0.06}$ | $97.55 \pm 19.67$ | $73.13 \pm 2.63$ | $76.30 \pm 2.63$ |
| RCAC (S) | $7.01 \pm 0.26$ | $\mathbf{93.64 \pm 21.23}$ | $\mathbf{70.13 \pm 2.25}$ | $76.10 \pm 1.92$ |

Table 11: Generalization performance in time for exact solving on SC, MIS, CA and CFL. We bold the best results and color the second-best results in green on each dataset.

| Model | SC # Nodes ↓ | MIS # Nodes ↓ | CA # Nodes ↓ | CFL # Nodes ↓ |
|---|---|---|---|---|
| FSB | $288.5 \pm 9.2$ | $12.0 \pm 3.8$ | $209.1 \pm 31.7$ | $153.4 \pm 4.4$ |
| RPB | $339.3 \pm 18.3$ | $10375.7 \pm 513.1$ | $18737.7 \pm 227.6$ | $180.3 \pm 13.1$ |
| VHB | $651.9 \pm 19.2$ | $423.2 \pm 33.1$ | $7255.38 \pm 330.01$ | $365.26 \pm 16.98$ |
| tMDP | $3475.8 \pm 210.0$ | $11612.1 \pm 789.9$ | $16039.6 \pm 1028.4$ | $608.7 \pm 20.7$ |
| GGCN (H) | $560.1 \pm 9.6$ | $13350.6 \pm 861.2$ | $12557.1 \pm 567.3$ | $412.9 \pm 18.5$ |
| RCAC (H) | $628.3 \pm 159.4$ | $11365.0 \pm 1808.5$ | $11843.0 \pm 2835.4$ | $\mathbf{406.2 \pm 21.9}$ |
| GGCN (S) | $\mathbf{404.6 \pm 4.3}$ | $7923.0 \pm 941.2$ | $11789.9 \pm 1105.9$ | $427.9 \pm 14.4$ |
| RCAC (S) | $462.5 \pm 46.4$ | $\mathbf{7102.8 \pm 1009.1}$ | $\mathbf{10965.2 \pm 997.6}$ | $425.0 \pm 16.8$ |

Table 12: Geeneralization performance in the size of search tree for exact solving on SC, MIS, CA and CFL. Human heuristics and neural methods are above and under the line separately. We bold the best results and color the second-best results for *neural methods* in green on each dataset.

Finally, we also ablate the choice of RCAC against other offline RL algorithms. Here we compare RCAC with CQL (Kumar et al., 2020) on the larger testing set in Table 13 and 14. It can be seen that RCAC takes a consistent lead against CQL over all datasets, no matter trained on the sub-optimal dataset or the small dataset.

| Model | SC Time (s) ↓ | MIS Time (s) ↓ | CA Time(s) ↓ | CFL Time(s) ↓ |
|---|---|---|---|---|
| CQL (H) | $1.71 \pm 0.03$ | $4.48 \pm 0.09$ | $2.70 \pm 0.17$ | $31.23 \pm 1.49$ |
| RCAC (H) | $1.69 \pm 0.05$ | $4.26 \pm 0.06$ | $2.65 \pm 0.07$ | $\mathbf{30.28 \pm 0.79}$ |
| CQL (S) | $1.68 \pm 0.04$ | $4.10 \pm 0.27$ | $3.84 \pm 0.36$ | $31.80 \pm 1.69$ |
| RCAC (S) | $\mathbf{1.65 \pm 0.02}$ | $\mathbf{3.86 \pm 0.32}$ | $\mathbf{2.46 \pm 0.04}$ | $31.68 \pm 1.60$ |

Table 13: Ablative results between RCAC and CQL in time for exact solving on SC, MIS, CA and CFL. We bold the best results on each dataset.

| Model | SC # Nodes ↓ | MIS # Nodes ↓ | CA # Nodes ↓ | CFL # Nodes ↓ |
|---|---|---|---|---|
| CQL (H) | $58.2 \pm 4.0$ | $246.8 \pm 90.9$ | $592.7 \pm 86.6$ | $526.0 \pm 20.0$ |
| RCAC (H) | $53.9 \pm 6.8$ | $161.7 \pm 17.0$ | $503.9 \pm 5.9$ | $\mathbf{487.5 \pm 19.9}$ |
| CQL (S) | $51.3 \pm 1.4$ | $104.2 \pm 17.2$ | $995.7 \pm 680.8$ | $541.2 \pm 40.3$ |
| RCAC (S) | $\mathbf{41.8 \pm 1.6}$ | $\mathbf{88.7 \pm 10.1}$ | $\mathbf{421.8 \pm 4.0}$ | $531.0 \pm 27.0$ |

Table 14: Ablative results between RCAC and CQL in the size of search tree for exact solving on SC, MIS, CA and CFL. We bold the best results on each dataset.

# E    DISTRIBUTION OF REWARDS

Here we visualize the distribution of rewards as a function of branching steps in Figure 4.

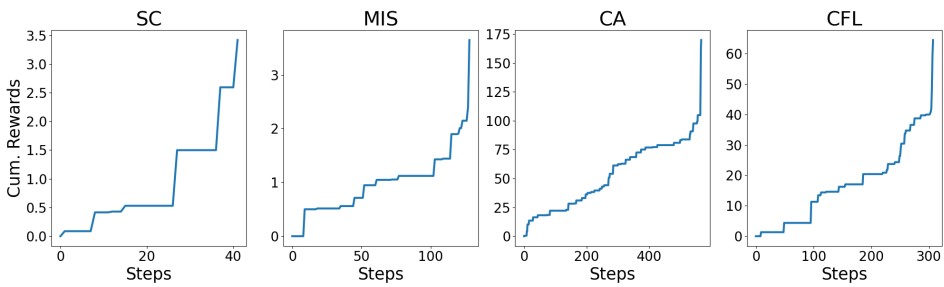

Figure 4: Cumulative rewards as a function of the branching steps.

