# OpenReview forum: "Learning to Branch with Offline Reinforcement Learning"
_ICLR.cc/2024/Conference — Submitted to ICLR 2024_

### Official Review · Reviewer_pc5v · 2023-10-29

**Soundness:** 2 fair
**Presentation:** 3 good
**Contribution:** 2 fair
**Rating:** 5
**Confidence:** 5

**Summary:**

This paper presents the Ranking-Constrained Actor-Critic algorithm, an offline reinforcement learning approach for optimizing Mixed Integer Linear Programs (MILPs). Traditional MILP solvers depend on hand-crafted heuristics for branching, limiting their efficiency and generalizability. Recent deep learning methods rely on high-quality training data, which can be scarce, particularly for large problems. The key contributions of the paper are the development of the new RL algorithm and its ability to efficiently learn branching strategies even from sub-optimal training data. The algorithm outperforms previous methods in terms of prediction accuracy and computational efficiency across various MILP problems, addressing the limitations of traditional solvers.

**Strengths:**

This paper claims to be innovative by being the first to apply offline reinforcement learning algorithms in branch-and-bound methods. Furthermore, the essence of the proposed method lies in further refining the dataset, specifically selecting the top-k actions in the set Gω for Bellman operator operations. This can effectively enhance the performance of the branching strategy. I believe this perspective can also be inspiring for similar problems in other domains.

**Weaknesses:**

This paper proposes training branch-and-bound strategies using offline reinforcement learning. However, in practice, interacting with solvers is relatively straightforward, and under these circumstances, using online reinforcement learning may yield better performance. The authors need to clarify the necessity of utilizing offline reinforcement learning.

**Questions:**

•	Considering that interacting with solvers online is convenient, is there a necessity to use offline reinforcement learning to train branch-and-bound strategies?
•	In Equation 7, when k is small, the distribution of Q-values over the dataset will be centered around -δ, which is unfavorable for training. How do the authors ensure training effectiveness in this scenario?
•	I believe that the essence of the method proposed by the authors lies in further refining the dataset, specifically selecting the top-k actions in Gω for Bellman operator operations. I am curious to know if, after obtaining the top-k actions in Gω, simple imitation learning on these state-action pairs would yield similar results as the current approach. In other words, my question is whether the key to the effectiveness of this algorithm lies in the dataset refinement rather than offline reinforcement learning. I suggest that the authors conduct further ablation experiments to validate this idea.

---

> ### Author Response · Authors · 2023-11-23
> **Response**
>
> We want to thank reviewer pc5v for your questions and constructive suggestions. Here are our responses to all of your comments.
>
>  > The authors need to clarify the necessity of utilizing offline reinforcement learning.
>
> We have briefly discussed the comparison in the third paragraph in our introduction part, here we will try to further explain why using offline reinforcement learning is meaningful.
>
> 1. Training reinforcement learning for branching is super time-consuming [1,2]. It is because interacting with the solver could also be slow (long interaction time) and the BnB tree could be super large when a RL policy is very bad at the beginning. Here we cite the training time from two recent RL for branching papers, `We set a maximum of 15,000 epochs and a time limit of six days for training` and `we left our retro branching agent to train for ≈ 13 days (≈ 500k epochs)`. In comparison, the offline models can all be trained within hours.
>
> 2. Bad performance of current RL methods. The results from the current two papers demonstrate that their performance is still way worse the imitation learning methods. In Table 2 and 3, we have also shown that the RL agent (tMDP) is even worse than RCAC which is trained over sub-optimal or small datasets.
>
> 3. There are lots of known problems in training online RL agents for branching, as summarized in [2]. For example, existing RL methods choose the size of the search tree as the reward function, which leads to a sparse reward environment and that is why we propose to use the improvement in the dual-bound as the reward function. A subsequent problem is the credit assignment problem (how to tell which action leads to the good/bad outcomes), which is studied by [1], but the the model performance shows that this problem is still far from being solved.
>
> [1] Lara Scavuzzo, Feng Yang Chen, Didier Chetelat, Maxime Gasse, Andrea Lodi, Neil Yorke-Smith, and Karen Aardal. Learning to branch with tree MDPs. In Alice H. Oh, Alekh Agarwal, Danielle Belgrave, and Kyunghyun Cho (eds.), Advances in Neural Information Processing Systems, 2022.
>
> [2] Christopher W. F. Parsonson, Alexandre Laterre, and Thomas D. Barrett. Reinforcement learning for branch-and-bound optimisation using retrospective trajectories. Proceedings of the AAAI Conference on Artificial Intelligence, 37(4):4061–4069, Jun. 2023.

---

> ### Author Response · Authors · 2023-11-23
> **Response (2)**
>
> > In Equation 7, when k is small, the distribution of Q-values over the dataset will be centered around -δ, which is unfavorable for training. How do the authors ensure training effectiveness in this scenario?
>
> Thanks for this great question. Actually, we do not directly fit the output of critic network $Q$ to $-\delta$. Note in Equation 8, we only change the output of the target network to $-\delta$. We realize this by first letting the target network output its value, then replace some of those outputs with $-\delta$. The target network needs no training, so it will not be fit to a distribution with values centered around $-\delta$. Then the fitting target of $Q$ in Equation 7 is also not a distribution with values centered around $-\delta$.
>
> > I believe that the essence of the method proposed by the authors lies in further refining the dataset, specifically selecting the top-k actions in Gω for Bellman operator operations.
>
> We think there could be some misconceptions here. $G_{\omega}$ is also a neural network rather than a set, so we actually do not do any dataset refining, i.e., filtering out good transitions from the dataset for training. The ranking is conducted on the scores of the $G_{\omega}$, so the top-k actions do not necessarily have to be in the collected dataset, that is, we do not observe the reward for all these top-k actions. But we can still evaluate these transitions with our learned Q-network.
>
> > I am curious to know if, after obtaining the top-k actions in Gω, simple imitation learning on these state-action pairs would yield similar results as the current approach. In other words, my question is whether the key to the effectiveness of this algorithm lies in the dataset refinement rather than offline reinforcement learning. I suggest that the authors conduct further ablation experiments to validate this idea.
>
> We believe the most relevant results are in Table 5, since $G_{\omega}$ has already been a neural network, simply imitating $G_{\omega}$ is just a distillation of the model. The better performance of RCAC over $G_{\omega}$ has already demonstrated that the source of improvement is from offline reinforcement learning.

---

### Official Review · Reviewer_s5Ux · 2023-10-29

**Soundness:** 3 good
**Presentation:** 2 fair
**Contribution:** 2 fair
**Rating:** 3
**Confidence:** 4

**Summary:**

This work proposes the usage of offline reinforcement learning for variable selection in the branch-and-bound algorithm. To do so, they introduce a novel offline algorithm that uses a classifier to determine whether a state-action pair is in the offline dataset. Their offline Q-values are now restricted towards picking only the top-k most likely actions for each state.

**Strengths:**

The usage of offline reinforcement learning seems more fitting than current imitation learning algorithms due to its lack of reliance on high quality demonstrations.

**Weaknesses:**

- The paper is a little unclear at some points. For instance, in the last paragraph of Section 2.2: Which variables are the selected ones? Just from the node chosen by the node selection policy, or all variables across the entire tree? In general, the distinction between node selection and variable selection doesn’t become clear: Does the method also do node selection (by picking variables from the entire tree), or just variable selection?
- Further, it is not exactly clear whether there is a single model trained and evaluated on all instances, or multiple independent models trained on and evaluated on individual datasets.
- One missing benchmark is the utilization of an off-the-shelf offline RL algorithm, such as conservative Q-learning as a baseline for the specific utility of RCAC over more established offline-RL algorithms (I.e. is the improvement in performance due to offline-RL or RCAC specifically?).
- The testing set is also rather small: 10k training instances, 2k validation instances and, 20 test instances is a strange ratio.
- The reward function is also a little bit strange: Why consider the dual bound, but ignore the primal one completely? Further, these bounds are not scale-invariant, meaning that the same problem, modulo a constant scalar, could have different dual bound improvements. Even if one takes care to normalize the objective vector c beforehand, most solvers like SCIP rescale this vector for increased numerical stability. Depending on which problems are chosen, the range of rewards across different instances might also be massive depending on the duality gap. However, we agree with the authors that this metric is still better than tree-size or number of nodes.

Some minor points:
- Abstract: hand-craft[ed]
- Intro: The sentence “All of these models are trained…” needs a re-write
- Intro: “To our knowledge, … to apply offline RL to MILP solving” (re-write)
- Sec. 2: typo pseudocsot
- Sec. 2.2. A[n] MDP
- Equation 4: one closing brace is too much (after $Q_\theta$)
- Sec. 3.1: when a[n] MILP instance
- Sec 3.1: discounted factor $\rightarrow$ discount factor
- Sec 3.3: citation of Gasse et al.: use cite instead of citep; same again happened in Sec. 4.1
- Sec. 4.1: please use cite and citep depending on how you add these citations into the text
- Sec. 5.2 does not add any benefit to the paper and can be omitted in its current state

**Questions:**

- Which set of variables if being selected from?
- What is the performance of other offline-RL algorithms?
- Can you evaluate on a larger testset?
- Why only look at the dual bound improvement (alternative: optimality gap between primal and dual)?
- In Sec 3.2. “In fact, a good action does no harm to policy optimization even if it is an OOD action” – can you please elaborate on this a bit more?

---

> ### Author Response · Authors · 2023-11-23
> **Response**
>
> We want to thank Reviewer s5Ux’s careful review and valuable advice. Some of the questions are actually related to the common practice in the existing work, but we will try our best to improve the experimental setting and clarify some descriptions based on your suggestions.
>
> > Which variables are the selected ones?
>
> Only the node chosen by the node selection policy is used. The node selection policy is fixed. Thanks for the question and we will make this part clear.
>
> > Further, it is not exactly clear whether there is a single model trained and evaluated on all instances, or multiple independent models trained on and evaluated on individual datasets.
>
> We have multiple independent models trained on and evaluated on individual datasets. Actually, the description of the experimental settings exactly follows the descriptions in the previous works, but we will add more clarifications in our final version.
>
> > One missing benchmark is the utilization of an off-the-shelf offline RL algorithm
>
> See Table 13 and 14 for the comparison with CQL.
>
> > The testing set is also rather small: 10k training instances, 2k validation instances, and 20 test instances is a strange ratio.
>
> This ratio is actually a common practice utilized in existing works. 20 test instances x 5 seeds leading to 100 measurements. We follow your suggestion to expand the number of testing instances to 100 in Table 9 and 10, which leads to 500 measurements. The reason for such a small testing set is due to its time measurement, which could be inaccurate under parallel computing, and we believe this is a common problem in the existing research.
>
> > Why only look at the dual bound improvement (alternative: optimality gap between primal and dual)?
>
> First, this is not a weird choice since the dual-integral is also a common metric to evaluate the quality of the branching policy, where only the dual bound is used. Second, dual-bound improvement is used to compute the score in the strong branching, which is empirically proven to be a powerful branching policy. It can somewhat directly reflect the quality of the instant branching decision leading to it while primal-dual gap cannot. Finally, the improvement from the primal bound is highly related to the primal heuristics, and is sometimes hard to fit with the node features.
>
>
> > In Sec 3.2. “In fact, a good action does no harm to policy optimization even if it is an OOD action” – can you please elaborate on this a bit more?
>
> Offline RL treats those actions with a low frequency in the dataset as OOD actions and avoids the query on these actions since the value estimation on these actions is inaccurate due to the lack of samples. However, if we can verify an action is good, for example, it brings a high dual-bound improvement, then we do not have to worry about its inaccurate value estimation. If its value is underestimated, this action will not be chosen by our actor, then there is no difference. If its value is over-estimated, our actor will choose this action, but since we have verified through the dual-bound improvement that it is highly possible to be a good action, it still brings no harm.

---

### Official Review · Reviewer_3oNR · 2023-10-31

**Soundness:** 2 fair
**Presentation:** 3 good
**Contribution:** 2 fair
**Rating:** 3
**Confidence:** 4

**Summary:**

This paper studies the problem of learning variable selection policies for mixed-integer linear programming (MILP). The authors propose an offline reinforcement learning (RL) approach to learn branching strategies from sub-optimal or inadequate training signals. Experiments demonstrate the proposed method outperforms baselines on various benchmarks.

**Strengths:**

1.	The paper is easy to follow.
2.	Experiments demonstrate the proposed method outperforms baselines on various benchmarks.

**Weaknesses:**

1.	The novelty of the proposed method is incremental, as the proposed method is a simple application of offline reinforcement learning methods to branching strategies learning.
2.	The authors claim that the proposed method is the first attempt to apply the offline RL algorithms to MILP solving. However, I found one previous work [1] applies offline RL methods to branching strategies learning as well.
3.	The authors may want to explain the novelty of their method over the work [1] in detail.
4.	The experiments are insufficient. First, the authors may want to evaluate their method on the load balancing dataset from the ML4CO competition as well. Second, the baselines are insufficient. The authors may want to compare their method to the work [1]. Third, the authors may want to evaluate the generalization ability of the learned models.

[1] Huang, Zeren, et al. "Branch Ranking for Efficient Mixed-Integer Programming via Offline Ranking-Based Policy Learning." Joint European Conference on Machine Learning and Knowledge Discovery in Databases. Cham: Springer Nature Switzerland, 2022.

**Questions:**

Please refer to Weaknesses for my questions.

---

> ### Author Response · Authors · 2023-11-23
> **Response (1)**
>
> We want to thank the reviewer’s questions, we have actually covered some of them in the paper, here we give more discussions about these points.
>
> > The novelty of the proposed method is incremental, as the proposed method is a simple application of offline reinforcement learning methods to branching strategies learning.
>
> RCAC is not just a simple application of one offline RL algorithm to learning to branch. We actually start with the CQL algorithm and gradually modify the offline RL algorithm to the current framework, here we summarize some core technical contributions in our work.
>
> Using a hard constraint (ranking) rather than the soft constraint adopted in many previous offline RL algorithms, such as CQL. This is because branching performance is somewhat sensitive to the branching decisions, a very bad branching decision would cause a huge subsequent BnB tree. Therefore, the offline RL agent should be trained more conservatively and that is why we use ranking as the constraint.
> Using a weighted cross entropy loss $G_{\omega}$ rather than a standard behavior cloning cross entropy loss. To overcome the over-conservativeness of RCAC, we give more preference to those promising actions when we rank the actions. This is realized through the dual-bound improvement, which is a heuristic adopted in strong branching.
>
> Using the dual-bound improvement as the reward function. As we discuss in section 3.1, compared with the BnB tree size, dual-bound improvement is denser and more flexible to compute, which serves as a better reward function.
>
> We also add the comparison with CQL in Table 13 and 14.

---

> > ### Author Response · Authors · 2023-11-23
> > **Response (2)**
> >
> > > The authors claim that the proposed method is the first attempt to apply the offline RL algorithms to MILP solving. However, I found one previous work [1] applies offline RL methods to branching strategies learning as well. The authors may want to explain the novelty of their method over the work [1] in detail.
> >
> > The difference between the mentioned paper and ours is actually discussed at the end of section 5.2. Though the name of the mentioned method is similar to our method, we want to point it out that these two methods are actually orthogonal.
> >
> > First, our assumptions are totally different. Our key assumption is that the interaction with the solver could be expensive, no matter through strong branching (Table 1) or the RL policy (the last part in our general response). This is also the basic assumption for current offline RL algorithms. Howeve, [1] assumes a super cheap cost of using strong branching. They not only collect the strong branching demos, but also use strong branching to roll out the trajectory at each step.
> >
> > Consequently, our research questions are also different. [1] solely focuses on improving the performance of neural branching, while we focus on the efficient training and learning from suboptimal data, which we believe is what existing offline RL algorithms do.
> >
> > Finally, two methods are actually orthogonal. [1] improves the algorithm for dataset collection while we improve the training algorithm. Though [1] claims they formulate the problem as an offline RL problem, they do not utilize any offline RL algorithm as we do. Therefore, we claim we are the first to `apply the offline RL algorithms to MILP solving`.
> >
> >
> > > First, the authors may want to evaluate their method on the load balancing dataset from the ML4CO competition as well. Third, the authors may want to evaluate the generalization ability of the learned models.
> >
> > The reason we do not evaluate our method on the load balancing dataset is because our assumption is that strong branching is costly to use and we do not have enough good demos. But load balancing is not in this scope, where thousands of strong branching demos could be easily collected in minutes. But we believe 4 datasets have already been enough given most previous works only use 4 or 5 datasets.
> >
> > > Second, the baselines are insufficient. The authors may want to compare their method to the work [1].
> >
> > As we discussed above, the methods are orthogonal and thus not comparable. [1] is innovative in their dataset collection algorithm while their training algorithm is still similar to imitation learning.
> >
> >
> > > Third, the authors may want to evaluate the generalization ability of the learned models.
> >
> > Thanks for the suggestion. See Table 11 and 12 in our updated version.
> >
> >
> > [1] Huang, Zeren, et al. "Branch Ranking for Efficient Mixed-Integer Programming via Offline Ranking-Based Policy Learning." Joint European Conference on Machine Learning and Knowledge Discovery in Databases. Cham: Springer Nature Switzerland, 2022.

---

### Official Review · Reviewer_nNZN · 2023-11-04

**Soundness:** 3 good
**Presentation:** 3 good
**Contribution:** 2 fair
**Rating:** 8
**Confidence:** 3

**Summary:**

The paper considers the problem of learning to select branching strategies while solving mixed integer programs via branch and bound algorithm. The key idea is to collect offline training dataset using full strong branching as behavior policy and learn an offline RL algorithm to generate the learned branching policy. Improvement of the dual bound is chosen as the reward function. Experiments are performed on four synthetic and two real world problems.

**Strengths:**

- Using offline RL for branching policies seems like a natural idea that should do better than pure imitation learning. I am surprised that this wasn't tried earlier and commend the paper for making this simple but natural idea work well.

- The description of the problem and solution is written clearly and easy to understand.

- The proposed approach performs well on multiple benchmarks.

**Weaknesses:**

- A large part of the paper talks about sub-optimality of the FSB policy. For example, this statement "Although FSB generally achieves high-quality branching, it could still become sub-optimal when the linear programming relaxation is uninformative or there exists dual degeneracy" Is there more justified argument for this backed by some evidence?

- why choose the proposed algorithm over any existing offline RL algorithm like CQL[1], IQL etc.?

[1] Kumar, A., Zhou, A., Tucker, G., & Levine, S. (2020). Conservative q-learning for offline reinforcement learning. Advances in Neural Information Processing Systems, 33, 1179-1191.

**Questions:**

- What are connections of equation 6 to reward weighed regression?

---

> ### Author Response · Authors · 2023-11-23
> **Response**
>
> We want to thank Reviewer nNZN for your positive feedback. Please see the responses below for your questions.
>
> > A large part of the paper talks about sub-optimality of the FSB policy. For example, this statement "Although FSB generally achieves high-quality branching, it could still become sub-optimal when the linear programming relaxation is uninformative or there exists dual degeneracy" Is there more justified argument for this backed by some evidence?
>
> The potential sub-optimality of Strong Branching has been pointed out in previous RL papers [1]. For example, in the multi-knapsack dataset, Strong Branching behaves clearly worse than RL methods. Even in Table 1, it can be observed that SB is worse than RPB.
>
> > why choose the proposed algorithm over any existing offline RL algorithm like CQL[1], IQL etc.?
>
> Using a hard constraint (ranking) rather than the soft constraint adopted in many previous offline RL algorithms, such as CQL. This is because branching performance is somewhat sensitive to the branching decisions, a very bad branching decision would cause a huge subsequent BnB tree. Therefore, the offline RL agent should be trained more conservatively and that is why we use ranking as the constraint.
>
> Please also see Table 13 and 14 for the comparison with CQL in our updated version.
>
> > What are connections of equation 6 to reward weighed regression?
>
> Equation 6 can be treated as a coarse-version of the reward weighed regression, since we only use the instant reward rather than the cumulative reward. Using instant rewards does not always make sense in the general RL formulation, but it works here since dual-bound improvement itself can reflect the quality of the branching decisions.
>
>
> [1] Lara Scavuzzo, Feng Yang Chen, Didier Chetelat, Maxime Gasse, Andrea Lodi, Neil Yorke-Smith, and Karen Aardal. Learning to branch with tree MDPs. In Alice H. Oh, Alekh Agarwal, Danielle Belgrave, and Kyunghyun Cho (eds.), Advances in Neural Information Processing Systems, 2022.

---

### Official Review · Reviewer_9gri · 2023-11-05

**Soundness:** 3 good
**Presentation:** 3 good
**Contribution:** 3 good
**Rating:** 5
**Confidence:** 4

**Summary:**

The authors propose an offline Reinforcement Learning (RL) framework for learning to branch (L2B) which reportedly exhibits superior performance with a sub-optimal dataset compared to existing methods that require extensive, high-quality datasets. This advantage is particularly notable in reducing the time to collect datasets for training the models. The reported performance on the MIP instances also indicates the effectiveness of the framework.

**Strengths:**

1. **Innovative Formulation:** The novel formulation of L2B as an Offline RL approach using a sub-optimal dataset is a significant departure from traditional methods.
2. **Efficiency in Data Collection:** The framework requires significantly less time to collect its dataset, enhancing its practicality.
3. **Performance:** The proposed framework improved performance compared to the GGCN framework on smaller dataset sizes, which is commendable.

**Weaknesses:**

Despite the novelty of the work, I have reservations about the robustness of its results. These concerns are expanded upon in this section and further detailed in the questions that follow.

1. **Lack of Scaling-Generalization Results:** A key aim of collecting datasets on smaller instances is to develop policies that excel on larger, more complex instances. It would be beneficial to see how various models perform on scaled-up versions of instances in various problem categories like SC, MIS, CA, or CFL. How do these policies perform on Medium or Hard instances (scaled-up versions) in SC, MIS, CA, or CFL? Does RCAC retain its performance advantage on scaling up to larger instances?

2. **Insufficient Comparison with Existing Methods:**
- The paper lacks a thorough comparison with recent advancements in the GGCN framework, particularly the augmented loss function introduced in "Lookback for Learning to Branch" (Gupta et al. 2022, https://arxiv.org/abs/2206.14987). It would be insightful to see how RCAC compares to this improved GGCN variant.
 - If I understand correctly, RCAC (S) and GGCN (S) primarily differ in their approach to training despite similarities in other aspects, such as dataset collection. Specifically, GGCN (S) employs a Cross-Entropy loss function, while RCAC (S) is focused on learning a Q-function (and a corresponding policy). The distinctiveness of the RCAC framework lies in its utilization of rewards instead of directly using FSB selections, as is the case with GGCN. However, an alternative comparison could involve integrating rewards into the GGCN framework as an additional signal. This could be achieved, for instance, by employing rewards to modulate the Cross-Entropy loss at each node, similar to how node depth might be used. Demonstrating RCAC's superior performance in this modified context would further reinforce the effectiveness of its RL-based approach as formulated in the study.
    - It would be valuable to have the values of \( k \) specified for each model. I am particularly curious to know whether \( k > 1 \) for RCAC(S).
- Comparisons with other RL methods, especially in terms of dataset size and time efficiency, would also be valuable.

**Questions:**

Clarifications:

1. **Section 3.3:** Should "representation of the B&B tree" be replaced with "representation of the B&B node" for accuracy?
2. **Training Dataset for GGCN (H) and RCAC (H):** Are these models trained on the same dataset? Is GGCN (H) trained on a separate dataset collected as specified in the Appendix?
3. **VHB Dataset Transitions:** Could the authors clarify what constitutes a 'transition' in this context? Does the transition include (s,a,s’) even when FSB is not employed in VHB, which is 0.05 times? Do you discard any transition? How is it ensured that you explore a wide array of instances before 100K transitions are collected?
4. **S Method Training:** Is the S method trained with only 5K transitions?
5. **Reward Distribution:** Could the authors provide details on the distribution of reward values in the dataset, perhaps in the Appendix? Information on how this varies with tree depth and how normalization is handled would be valuable.
6. **Figure 3 Clarity:** What is the specific problem family represented in Figure 3?
7. **Practicality of H dataset collection:** Given that VHB takes longer than FSB (as indicated in column 2), is it still a practical choice since the performance is worse than S?
8. **GGCN Expansion:** Could the authors clarify the abbreviation GGCN? It seems to be a variation of GCNN (Graph Convolutional Neural Networks) as used in Gasse et al. 2019.
9. **Inference Procedure in RCAC:** Are there two forward passes $G_\omega\$ and $\pi_\phi$ during inference in RCAC? How does this differ from the inference process in GGCN?
10. **Hyperparameter \(k\):** Figure 3 suggests that \(k\) has a significant impact on RCAC's performance. Could the authors provide the \(k\) values used for each model and dataset?

11. **Aggregation in Table 4:** How are scores aggregated across 20 instances in Table 4? Assuming this is a cumulative sum, RCAC appears to outperform in WA but not against RPB in AP. Can the authors speculate on which problem types might be more amenable to improvement by RCAC?

12. **Reward Ablation:** Could the authors discuss the rationale behind choosing dual bound improvement over primal-dual gap improvement? Understanding the preference for one metric over the other would be enlightening.


Suggestions:
1. **Dataset Comparison:** I think it will be pretty helpful to have a section or a figure demonstrating the difference (transition vs. individual nodes) between the dataset collected using the standard IL methods and the one proposed in this work.
2. **Statistical Significance:** Please include p-values to indicate the statistical significance of differences in Tables 2 and 3.
3. **Evaluation Methodology:** Given that 20 seems a relatively small sample size for testing, it's common practice to evaluate each instance with multiple seeds, as demonstrated in Gasse et al. 2019. Could the authors clarify whether a similar approach can be employed in their study?

---

> ### Author Response · Authors · 2023-11-23
> **Response (1)**
>
> We want to thank Reviewer 9gri for your questions and constructive suggestions on our empirical evaluations. Please see the following responses for your questions.
>
> > Lack of Scaling-Generalization Results:
>
> See Table 11 and 12 in our updated version
>
>
> > The paper lacks a thorough comparison with recent advancements in the GGCN framework, particularly the augmented loss function introduced in "Lookback for Learning to Branch".
>
> This paper focuses more on a better imitation of the strong branching, which is a bit different from what we study. Besides, the code for this paper is not public, we will try to ask the authors for the code or reproduce it in the final version.
>
> > However, an alternative comparison could involve integrating rewards into the GGCN framework as an additional signal.
>
> Actually, $G_{\omega}$ is trained in the way you suggest. Our result in Table 5 indicates that $G_{\omega}$ has already brought some improvements, but RCAC can further boost the performance. Also, to ensure the reward if informative, we choose the dual-bound rather than the tree size used in previous RL for branching methods.
>
> > Comparisons with other RL methods, especially in terms of dataset size and time efficiency, would also be valuable.
>
> We actually discuss it in Section 4.2. Here we cite the training time from two recent RL for branching papers, `We set a maximum of 15,000 epochs and a time limit of six days for training` [1] and `we left our retro branching agent to train for ≈ 13 days (≈ 500k epochs)` [2] . In comparison, the offline models can all be trained within hours.
>
>
> [1] Lara Scavuzzo, Feng Yang Chen, Didier Chetelat, Maxime Gasse, Andrea Lodi, Neil Yorke-Smith, and Karen Aardal. Learning to branch with tree MDPs. In Alice H. Oh, Alekh Agarwal, Danielle Belgrave, and Kyunghyun Cho (eds.), Advances in Neural Information Processing Systems, 2022.
>
> [2] Christopher W. F. Parsonson, Alexandre Laterre, and Thomas D. Barrett. Reinforcement learning for branch-and-bound optimisation using retrospective trajectories. Proceedings of the AAAI Conference on Artificial Intelligence, 37(4):4061–4069, Jun. 2023.
>
>
>
> > Section 3.3: Should "representation of the B&B tree" be replaced with "representation of the B&B node" for accuracy?
>
> Yes, thanks for the suggestion and we have updated it.
>
> > Training Dataset for GGCN (H) and RCAC (H): Are these models trained on the same dataset? Is GGCN (H) trained on a separate dataset collected as specified in the Appendix?
>
> Yes. Please see Section 4.1, the paragraph above Table 1, for more details about the two datasets.
>
> > VHB Dataset Transitions: Could the authors clarify what constitutes a 'transition' in this context? Does the transition include (s,a,s’) even when FSB is not employed in VHB, which is 0.05 times? Do you discard any transition? How is it ensured that you explore a wide array of instances before 100K transitions are collected?
>
> Each transition, as defined in section 2.3, includes (s,a,s’, r(s,a)), which corresponds to the current state, current action, next state and the instant reward. It is always the same no matter VHB or SB is used. We do not discard any transition or make any sub-sequence sampling due to our basic assumption that collecting the samples is expensive. So the exploration of diverse instances is not considered in our case.
>
> > S Method Training: Is the S method trained with only 5K transitions?
>
> Yes.
>
> > Reward Distribution: Could the authors provide details on the distribution of reward values in the dataset, perhaps in the Appendix?
>
> See Figure 4 in our updated version.
>
> > Figure 3 Clarity: What is the specific problem family represented in Figure 3?
>
> As the title suggests, it is the CA (combinatorial auction) problem. We change the ranking constraint $k$ and report of performance change of RCAC.
>
> > Practicality of H dataset collection: Given that VHB takes longer than FSB (as indicated in column 2), is it still a practical choice since the performance is worse than S?
>
> The answer could also be found in the paragraph we refer to in the response to clarification 2. In short, VHB is used to simulate the scenario where the best available heuristics are sub-optimal. Since on these 4 commonly used datasets, SB has already been strong enough, we use VHB to simulate a sub-optimal policy and evaluate the performance of RCAC. Besides, models also have a better performance when trained on the dataset collected by VHB on the CFL problem, so it could also sometimes become a better choice.
>
> > GGCN Expansion: Could the authors clarify the abbreviation GGCN? It seems to be a variation of GCNN (Graph Convolutional Neural Networks) as used in Gasse et al. 2019.
>
> We actually use GGCN to refer to the method used in Gasse et al. 2019. Since GCN or GCNN is more like a model name, which is also adopted in our learning method, to discriminate the learning method name from the model name, we try to use the author name G(asse)GCN to name the method.

---

> > ### Author Response · Authors · 2023-11-23
> > **Response (2)**
> >
> > > Inference Procedure in RCAC
> >
> > Only the actor is used during the inference in the current results, as we state in Section 3.2, `During inference, the action arg maxa $\pi_{\phi}(a|s)$ is used at each step. Our algorithm could be summarized as follows`. Note that the actor has already been trained over the Q-values in Equation 7 to avoid those OOD actions, so $G_{\omega}$  is no longer needed. However, we actually observe some improvements when $G_{\omega}$ is used udring the inference time. Theoretically, the inference time remains unchanged if $\pi$ and $G_{\omega}$ are run in parallel, we may add this result in the final version.
> >
> > > Hyperparameter (k)
> >
> > See Table 7 in our updated version.
> >
> >
> > > How are scores aggregated across 20 instances in Table 4?
> >
> > This is actually the average reward across 20 instances (though the number looks quite large), details about the reward can be found in https://www.ecole.ai/2021/ml4co-competition/ (in one word, negated unshifted dual integral).
> >
> > > RCAC appears to outperform in WA but not against RPB in AP. Can the authors speculate on which problem types might be more amenable to improvement by RCAC?
> >
> > Anonymous Problem (AP) is a heterogeneous dataset where problems have different types (for example, it could be a mixture of SC, MIS, CA, CFL and other problems as well). The generalization of such a dataset could be hard since different problems may have different preferences for branching. RCAC can bring improvement to the dataset with a clear distribution.
> >
> > > Reward Ablation: Could the authors discuss the rationale behind choosing dual-bound improvement over primal-dual gap improvement? Understanding the preference for one metric over the other would be enlightening.
> >
> > Dual-bound improvement is used to compute the score in the strong branching, which is empirically proven to be a powerful branching policy. It can somewhat directly reflect the quality of the instant branching decision leading to it while the primal-dual gap cannot. Besides, the improvement from the primal bound is highly related to the primal heuristics and is sometimes hard to fit with the node features.
> >
> > > Dataset Comparison: I think it will be pretty helpful to have a section or a figure demonstrating the difference (transition vs. individual nodes) between the dataset collected using the standard IL methods and the one proposed in this work.
> >
> > Thanks for the suggestion. Due to the page limit, we will consider adding such a Figure in our final version.
> >
> > > Statistical Significance: Please include p-values to indicate the statistical significance of differences in Tables 2 and 3.
> >
> > See Table 8 in our updated version.
> >
> > > Evaluation Methodology: Given that 20 seems a relatively small sample size for testing, it's common practice to evaluate each instance with multiple seeds, as demonstrated in Gasse et al. 2019. Could the authors clarify whether a similar approach can be employed in their study?
> >
> > Yes. We also increased the number of testing instances to 100 in Tables 9 and 10 in the updated version.

---

### Author Response · Authors · 2023-11-23
**General Response**

We want to first thank reviewers for recognizing the novelty of our work in applying offline RL to learn from sub-optimal branching demonstrations, and we value the constructive feedbacks from all reviewers. To summarize the update on our paper, we add Table 7 to detail the choice of hyperparameter k (Reviewer 9gri); Table 8 to show the p-values for Table 2 and 3 (Reviewer 9gri); Table 9 and 10 as the performance on a larger testing set (Reviewer s5UX); Table 11 and 12 to include the generalization performance (Reviewer 9gri, 3oNR); Table 13 and 14 to compare with CQL (Reviewer nNZN, s5UX); and Figure 4 to visualize the distribution of rewards vs. the number of branching steps (Reviewer 9gri).

We have highlighted all changes in the blue color. Due to the time and resource limit, we will try to incorporate the remaining suggestions in our final version. We also respond to other questions from each reviewer in the individual response.

---

### Meta-Review · Area_Chair_PxXq · 2023-12-09

**Metareview:**

This paper considers improving branch-and-bound approaches for solving MILP problems. It learns branching strategies from offline training data collected from full strong branching using an offline reinforcement learning approach.

The most positive aspect of the paper is that it takes a natural approach to solve this problem which is not explored before. However, the reviewers' had some outstanding concerns during the discussion:
- Differentiating this approach both qualitatively and quantitatively with Huang, Zeren, et al. "Branch Ranking for Efficient Mixed-Integer Programming via Offline Ranking-Based Policy Learning." Joint European Conference on Machine Learning and Knowledge Discovery in Databases, 2022.
- Experiments on load balancing dataset from the ML4CO competition requested by one of the reviewer were not provided.
- Clarifying the assumption of having only low-quality demonstration data without wanting to collect better-quality data online in interaction with the solver really holds in practice or not.
- Explaining the results for GGCN and RPB in Table 11 and Table 12 which are inconsistent with prior work.

Therefore, I recommend rejecting the paper and strongly encourage the authors' to revise the paper based on the review comments for re-submission.

**Justification For Why Not Higher Score:**

Significant weaknesses as mentioned in the meta review.

**Justification For Why Not Lower Score:**

N/A

---

### Decision · Program_Chairs · 2024-01-16

Reject